# Automatic Segmentation of Specific Intervertebral Discs through a Two-Stage MultiResUNet Model

**DOI:** 10.3390/jcm10204760

**Published:** 2021-10-17

**Authors:** Yu-Kai Cheng, Chih-Lung Lin, Yi-Chi Huang, Jui-Chi Chen, Tzu-Peng Lan, Zhen-You Lian, Cheng-Hung Chuang

**Affiliations:** 1Department of Neurosurgery, China Medical University Hospital, Taichung 404, Taiwan; master3743@gmail.com; 2Department of Neurosurgery, Asia University Hospital, Taichung 413, Taiwan; jefflin0529@gmail.com; 3Department of Occupational Therapy, Asia University, Taichung 413, Taiwan; 4Department of Radiology, Asia University Hospital, Taichung 413, Taiwan; yichi710103@gmail.com; 5Department of Computer Science and Information Engineering, Asia University, Taichung 413, Taiwan; rikki@asia.edu.tw (J.-C.C.); zzzlarry.zp@gmail.com (T.-P.L.); s00098122@gmail.com (Z.-Y.L.)

**Keywords:** deep learning, U-Net, spine image, degenerative disc, intervertebral disc segmentation

## Abstract

The automatic segmentation of intervertebral discs from medical images is an important task for an intelligent clinical system. In this study, a deep learning model based on the MultiResUNet model for the automatic segmentation of specific intervertebral discs is presented. MultiResUNet can easily segment all intervertebral discs in MRI images; however, when only certain specific intervertebral discs need to be segmented, problems with segmentation errors, misalignment, and noise occur. In order to solve these problems, a two-stage MultiResUNet model is proposed. Connected-component labeling, automatic cropping, and distance transform are used in the proposed method. The experimental results show that the segmentation errors and misalignments of specific intervertebral discs are greatly reduced, and the segmentation accuracy is increased to about 94%. The performance of the proposed method proves its usefulness for the automatic segmentation of specific intervertebral discs over other deep learning models, such as the U-Net, CNN-based, Attention U-Net, and MultiResUNet models.

## 1. Introduction

Back pain is very common and a major public health problem that affects adults of all ages [1,2]. It often brings pain and distress to patients and leads to a decline in the quality of life. The literature has shown that the point prevalence in a number of studies ranged from 12% to 35%, and around 10% of sufferers became chronically disabled [2]. Back pain also places an enormous economic burden on society and its costs include direct medical costs, insurance, lost production, and disability benefits [2]. Degeneration of intervertebral discs is one of the causes of back pain. Intervertebral disc degeneration, which often causes back pain, is a natural aging process characterized by changes in the appearance and chemical structure of the intervertebral discs [2].

The spine is composed of cervical vertebrae (C1–C7), thoracic vertebrae (T1–T12), lumbar vertebrae (L1–L5), and sacral vertebrae (S1–S5) [3] and its main function is to bear the weight of the body. The lumbar spine consists of five vertebrae (L1–L5). These five vertebrae are the largest vertebrae in the spine, and they are strong enough to support most of the weight of the body [4]. Spinal degeneration usually starts from the lumbar spine, because the lumbar spine bears most of the weight of the entire upper body. Clinically, physicians hope to analyze and diagnose differences between normal and degenerated spines by measuring the height of the lumbar intervertebral discs [4,5]. For example, physicians usually want to compare the height and state of the intervertebral discs of the lumbar spine at L1/L2, L4/L5, and L5/S1 for age-related spinal degeneration, as the intervertebral discs at L4/L5 and L5/S1 are more prone to spinal degeneration, while the intervertebral disc at L1/L2 is less prone to degeneration [6]. If such a workflow is completed manually, it is laborious and time-consuming. Therefore, an automatic segmentation and recognition method for specific intervertebral discs, but not all of them, is required.

Imaging methods are the most important diagnostic modalities in degenerative spinal disease [7]. Computed tomography (CT) or magnetic resonance imaging (MRI) are often clinically used to obtain images of the spine for diagnosing spine-related diseases, such as lower extremity radiating pain, intervertebral disc herniation, lumbar spinal stenosis, and spinal degeneration. Between them, the texture of MRI images can better reflect the characteristics of biochemical and structural tissues [8]. MRI can also detect the fat and moisture content of the intervertebral discs and vertebral segments. Moreover, the use of MRI to diagnose patients can usually obtain a higher diagnosis accuracy rate, which grants it greater clinical application value. Therefore, this study focuses on MRI imaging as the experimental material.

Recently, convolutional neural networks (CNN) have brought breakthroughs in the field of image segmentation and recognition, especially for medical images. This has made the application of deep learning in the medical field increasingly extensive. Ronneberger et al. [9] proposed the U-Net model to use available annotated samples more efficiently in the segmentation of neuronal structures in electron microscopic stacks. Kayalibay et al. [10] proposed a CNN-based method with three-dimensional filters and applied it to segment hand and brain MRIs. Oktai et al. [11] proposed an attention U-Net model that can be used for medical-imaging segmentation and can automatically learn to focus on target structures of varying shapes and sizes. Ibtehaz et al. [12] proposed the MultiResUNet model to improve on the U-Net model and to segment multimodal medical images. Lou et al. [13] designed a DC-UNet model modified from U-Net and obtained a relative improvement in performance, compared with classical U-Net.

Various deep learning models have been proposed in the literature to segment MRI images of intervertebral discs [14,15,16]. Wang et al. [14] proposed a convolutional framework, based on 3D U-Net, to segment 66 intervertebral discs from multimodal MRI images and obtained an average Dice coefficient of 89.0%, i.e., an intersection over the union (IoU) of 80.2%. Vania et al. [15] developed a multistage optimization mask-RCNN for the intervertebral disc-instance segmentation of 263 patients with T1 and T2 images and acquired an average Dice coefficient of 99.5% (IoU of 98.9%). Das et al. [16] introduced a region-to-image matching network model to identify and segment intervertebral discs in 24 multimodal MRI images from 16 subjects and obtained an average identification accuracy of 92.5% and a segmentation Dice coefficient of 91.7% (IoU of 84.7%). However, most of these methods segment all the intervertebral discs in an image instead of segmenting specific discs. Furthermore, they have to address the problem of having little original image data.

In this study, we used a dataset of about spinal 3000 MRI images. We used several deep learning models to segment three specific intervertebral discs at L1/L2, L4/L5, and L5/S1. We found that some incorrect segmentation existed, and the average IoU values were all lower than 72.3%. These incorrect segmentation results included the segmentation of redundant intervertebral discs or the segmentation of wrong (non-specific) discs. Simultaneously, we found that MultiResUNet superior to other models. Therefore, we propose a two-stage method based on MultiResUNet to improve the accuracy of segmentation. The experimental results show that the segmentation errors and misalignments of intervertebral discs were reduced, and the accuracy of correct segmentation increased, to about 94%.

## 2. Materials

According to the estimation of sample size for confidence interval estimates [17], if a 95% confidence interval, a standard deviation of 0.01, and a margin of error of 0.04% are used, the most conservative sample size would be about 2400. A total of 2982 de-identified spine lateral images obtained between August 2016 and July 2020 at Asia University Hospital, Taichung, Taiwan, were collected. This number exceeds the most conservative sample size required. The slice of the midsagittal section of lumbar vertebrae was selected to be the image data. The acquired spine lateral images came from MRI scanners, primarily including the thoracic, lumbar, and sacral vertebrae, and were transferred to 512 × 512 bitmap images. These images were manually marked (three intervertebral discs between the lumbar spine and the sacrum) by a physician, i.e., L1/L2, L4/L5, and L5/S1. In the collected images, cases of anatomical abnormalities that lacked these three intervertebral discs were removed, such as lumbar sacrization or sacrum lumbarization. As the number of images in these cases was small and this study is focused on the segmentation and measurement of intervertebral discs, we did not classify any spinal diseases in the image data. However, we wish to measure the size and inhomogeneity of the segmented intervertebral discs as a study of the correlation between intervertebral discs and spinal degeneration in the future. As long as the three intervertebral discs (L1/L2, L4/L5, and L5/S1) of the image could be manually segmented, it was selected for inclusion in our experimental data. The collected images were randomly divided into a training set of 2674 images and a testing set of 308 images. Figure 1 shows an example of an original MRI image and its corresponding standard masks of the three intervertebral discs under examination.

## 3. Methods

### 3.1. MultiResUNet

In this study, we first used the MultiResUNet model to segment three specific intervertebral discs. The MultiResUNet model can easily implement the training work of segmentation of these three specific intervertebral discs; however, segmentation errors often occur during the testing phase. These errors include the segmentation of redundant intervertebral discs or the segmentation of the wrong (non-specific) intervertebral discs. Figure 2 shows an example of a segmentation error, where an extra intervertebral disc is segmented. In order to solve these segmentation error problems, this study proposes a two-stage method based on the MultiResUNet model. For more technical description of MultiResUNet, please refer to the Appendix A.

### 3.2. The Two-Stage Method

In the two-stage method, the lower two intervertebral discs are segmented in the first stage, and then the upper intervertebral disc is trained for segmenting using a distance feature obtained in the second stage. In our method, specific intervertebral discs must be assigned different labels. Connected-component labeling is applied to relabel the training data so that each disc has a different label. In our implementation, the background was marked as zero, and the three specific intervertebral discs were marked as one to three, from top to bottom. Figure 3 shows the result of the connected-component labeling for the three specific intervertebral discs. We cut out the lower two intervertebral discs from the training data in the first stage. Figure 4 shows the cropped images that were used to train the Multi-ResUNet in the first stage. In this phase, only the lower two intervertebral discs were segmented by the MultiResUNet model.

The segmentation of the upper intervertebral disc (labeled as 1) was achieved by the MultiResUNet model in the second stage. As the L1/L2 intervertebral discs are very similar to their adjacent intervertebral discs, they have a high probability of segmentation error. Therefore, in the second stage, distance features were added to modify the training data to help with identification. A distance transform was used to compute the distance features of the lower two intervertebral discs. The distance value gradually expanded outward from zero at the lower two intervertebral disc areas. Then, the original MRI image, combined with these distance maps, was used to train the MultiResUNet to segment the upper intervertebral disc. Figure 5 shows the training data of the original MRI image combined with the distance maps. In this step, only the L1/L2 intervertebral discs were segmented by the MultiResUNet model. For more technical description of the two-stage method, please refer to the Appendix B.

## 4. Results

The experimental equipment is shown as follows. The hardware used was an Intel^®^ Core™ i7-8700 processor and 32GB memory, with a NVIDIA GeForce GTX 1070 8GB graphics card. The software used was Windows 10 64-bit operating system, CUDA Toolkit 11.1.0 and cuDNN 10.1. The deep learning framework was PyTorch1.8.1+cu111+Python3.7.4. 

### 4.1. Evaluation Metrics

In order to evaluate the effectiveness of the methods, the intersection over union (IoU) [9,12], which is also known as the Jaccard index, was used. IoU is calculated by the ratio of the intersection and union of two sets. Assuming that the two sets are X and Y, X is the standard mask and Y is the segmented area; formula is as follows:(1)$$\mathrm{IoU}=J(X,Y)=\frac{X\cap Y}{X\cup Y}.$$

In addition, in order to judge whether the segmentation of the intervertebral disc is correct or not, this study set discrimination criterion. Assuming that the total number of images in the testing set is *T*; first, the number of segmented intervertebral discs in the *i*-th predicted image is computed and marked as *P_i_* by using connected-component labeling. The standard number of intervertebral discs, marked as *L*, was set to three in this study. If the number of segmented intervertebral discs in a predicted image was not equal to the standard value *L*, then this predicted image was regarded as an error prediction.

After removing the predicted images with larger or less than the standard value *L*, the intersection area of the standard mask and the corresponding segmented region in the predicted image was calculated. The ratio of this intersection area to the area of the standard mask was the discrimination criterion. Assuming that the total number of images in the testing set was *T*, the discrimination criterion, the areas of the predicted result, and the corresponding standard mask of the *k*-th intervertebral disc in the *i*-th predicted image were *C_ik_*, *R_ik_*, and *S_ik_*, respectively, where *k* = 1, 2, 3 represents the three intervertebral discs. The formula of the discrimination criterion is as follows: (2)$$C_{i}^{k}=C(R_{i}^{k},S_{i}^{k})=\frac{R_{i}^{k}\cap S_{i}^{k}}{S_{i}^{k}}.$$
where *i* = 1, 2, …, *T* and *k* = 1, 2, 3. If the discrimination criterion of an intervertebral disc in the predicted image is less than 70%, then this predicted image is regarded as an error prediction. 

Next, a binary value, denoted as *b_i_*, is used to represent the predicted image as follows: (3)$$b_{i}=\{\begin{array}{l} 1,\;\mathrm{where}\;P_{i}=L\;\mathrm{and}\;C_{i}^{k}\geq 70\%\\ 0,\;\mathrm{otherwise}\; \end{array},$$
where *i* = 1, 2, …, *T* and *k* = 1, 2, 3. When the predicted image is correct, *b_i_* is marked as 1, and when the predicted image is incorrect, *b_i_* is marked as 0. Assuming that the prediction accuracy is *A* and *B* represents the number of correctly predicted images, accuracy *A* is equal to *B* divided by *T*. The formula of the accuracy is calculated as follows:(4)$$A=\frac{B}{T}=\frac{1}{T}\sum_{i=1}^{T}b_{i}.$$

### 4.2. Performance Comparison and Discussion 

In order to compare the performance of different methods, five different models were tested in this experiment, namely U-Net [9], CNN-based [10], Attention U-Net [11], MultiResUNet [12], and the proposed two-stage MultiResUNet model. The testing images to be evaluated as the correct prediction needed to satisfy two conditions. The number of segmented intervertebral discs was equal to three and the discrimination criteria of segmented intervertebral discs in the predicted image were all greater than 70%. The comparison of the proposed method and the other existing models is summarized in Table 1. As the testing set contained 308 images, i.e., T = 308, accuracy, *A,* is equal to the number of correctly predicted images, *B,* divided by 308. Regarding the calculation of mean IoU, all the IoU values of the segmentation results of the 308 testing images needed to be summed and averaged. Although all the testing images had standard masks to evaluate whether the segmentation was correct, the correct segmentation results should not be the sole calculation when computing the mean IoU. The comparison results showed that the accuracy and mean IoU of the proposed method were 93.8% and 77.1%, respectively, which were better than for the other four models. 

In our experiments, focal loss [18] was represented loss function in all models. Figure 6 shows the loss curve of each model during training. Due to the different input training data in the second stage of the proposed method, the loss curve of the second stage is different from the others; however, these models are all able to successfully converge within about 200 epochs.

Next, we observed the variation in IoU of each model during training and validation. During the training phase, the IoU values of all models quickly exceeded 70% and continued to rise until exceeding 80%. However, during the validation phase, the IoU values of all models fluctuated around 60% to 70%, as epoch increased. The main reason for this was segmentation errors, which prevented the IoU values from increasing. Figure 7 shows the IoU curves during the training and validation processes of each model. Although the IoU of our proposed method was not as successful as the other four models during the training phase, it perfomed best in the validation process.

Finally, we compared the time cost during the training phase of each model, as shown in Figure 8. The CNN-based model took the least training time; it took about 28,388 s, i.e., 7 h, 53 min and 8 s. The Attention U-Net model took the longest time; about 290,798 s, i.e., 3 days, 8 h, 46 min and 38 s. Since the proposed method performed two training processes in the first stage and the second stage, it took 130,660 s in parallel processing, i.e., 1 day, 12 h, 17 min, and 40 s. It took less time than the Attention U-Net, U-Net and MultiResUNet models, if we trained the first and second stages at the same time. The advantage of the proposed method is that parallel processing can be used in the training process, which the other models cannot support. The advantage of the U-Net-based methods is that the training process is fast and the training of a small amount of data can produce good segmentation performance.

To sum up, the proposed method effectively achieved the segmentation of specific intervertebral discs at L1/L2, L4/L5, and L5/S1. Its accuracy and average IoU values were better than the U-Net, CNN-based, Attention U-Net, and MultiResUNet models. During the training process, loss in all five models was easy to drop and converge. In the validation process, the IoU of the proposed method approached 80% and was better than the other four models. In terms of time costs, the proposed method can adopt parallel processing, and ranks second among the five models.

## 5. Conclusions

This study proposed a two-stage MultiResUNet model to segment required specific intervertebral discs for clinical diagnosis. In the first stage, the method uses a small range of MRI images partially segment the target and then extracts a distance feature, using the distance transform from the initial segmentation result. In the second stage, the original MRI image is combined with these distance maps and is used to train a MultiResUNet algorithm to segment the remaining targets. The experimental results show that segmentation performance was significantly improved by our proposed model. In the evaluation, the segmentation accuracy was about 94% and mean IoU was about 77%, an increase of about 10% and 5%, respectively, compared with the results from the MultiResUNet model. MultiResUNet was used as the main network architecture in the proposed method. Although good results have been achieved, there is still a long way to go before its use in practical application is viable. However, given the excellent structure of the U-Net model, many excellent variants have emerged, such as ResUNet-a [19] and DC-UNet [13]. Further research directions include using a better model than MultiResUNet, adding more training data, or using a loss function that is more suitable for intervertebral disc segmentation in order to improve the accuracy of the segmentation. The limitation of this method is that it cannot be used in patients who lack intervertebral discs at L1/L2, L4/L5, and L5/S1. In clinical applications, the proposed method can facilitate the analysis and comparison of specific intervertebral discs by neurosurgeons and save the cost of manual segmentation. In addition, this method can also provide subsequent measurement of the size and inhomogeneity of intervertebral discs for more clinical research in the future, for example, in studying the correlation between intervertebral discs and spinal degeneration.

## Figures and Tables

**Figure 1 jcm-10-04760-f001:**
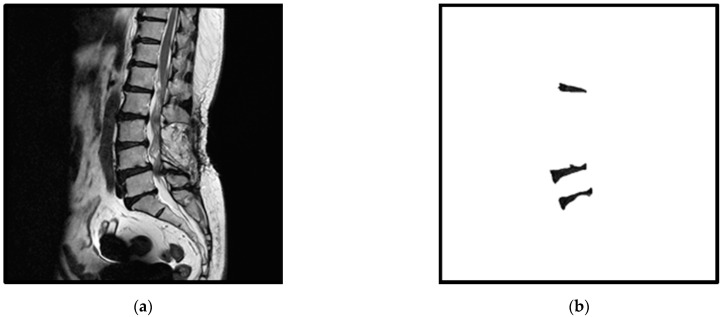
An example of an original spine MRI image and its corresponding standard masks of the three considered intervertebral discs: (**a**) the original spine MRI image; (**b**) the corresponding standard masks of the three intervertebral discs of (**a**).

**Figure 2 jcm-10-04760-f002:**
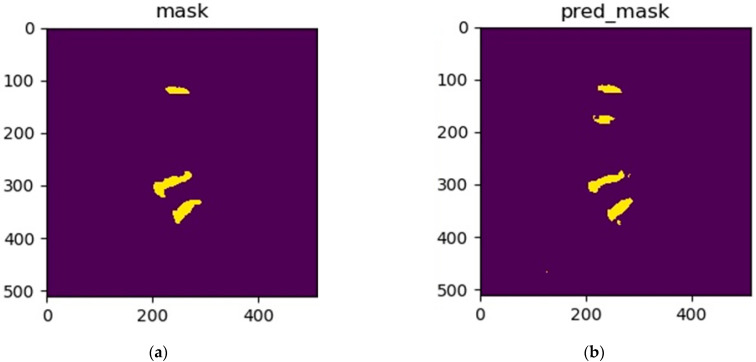
An example of a segmentation error: (**a**) the standard masks of the three intervertebral discs; (**b**) the segmentation error of a redundant intervertebral disc using the MultiResUNet model.

**Figure 3 jcm-10-04760-f003:**
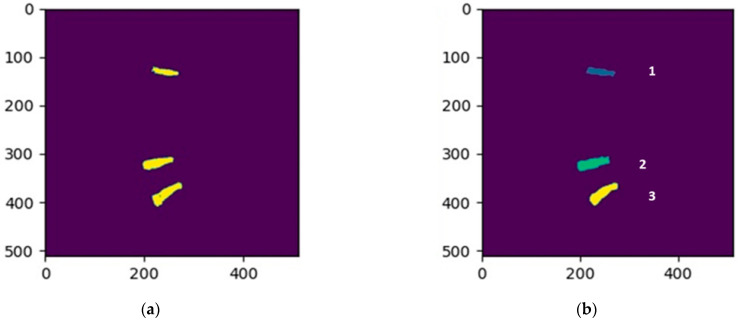
The sampling result of connected-component labeling for the three specific intervertebral discs: (**a**) the standard masks of the three intervertebral discs; (**b**) the connected-component labeling result of (**a**).

**Figure 4 jcm-10-04760-f004:**
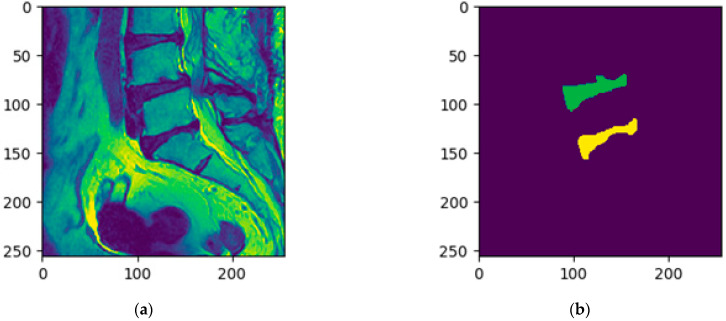
The cropped images that were used to train the MultiResUNet in the first stage: (**a**) the cropped original spine MRI image (shown in the viridis color maps); (**b**) the corresponding standard masks of the lower two intervertebral discs of (**a**).

**Figure 5 jcm-10-04760-f005:**
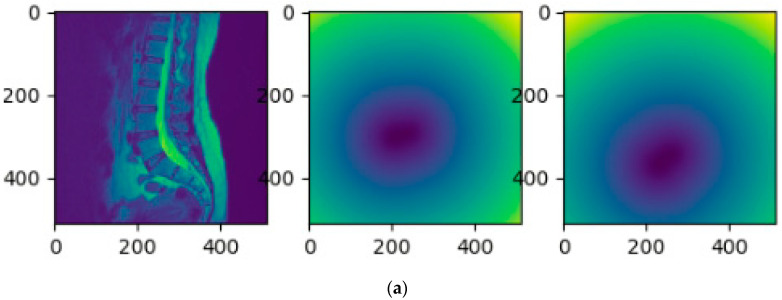
The training data of the second stage is a combination of distance maps and the original MRI image: (**a**) the original MRI image and the distance maps of the lower two intervertebral discs (shown in the viridis color maps); (**b**) the original MRI image combined with the distance maps (shown in RGB color maps); (**c**) the corresponding standard mask of the L1/L2 intervertebral discs of (**b**).

**Figure 6 jcm-10-04760-f006:**
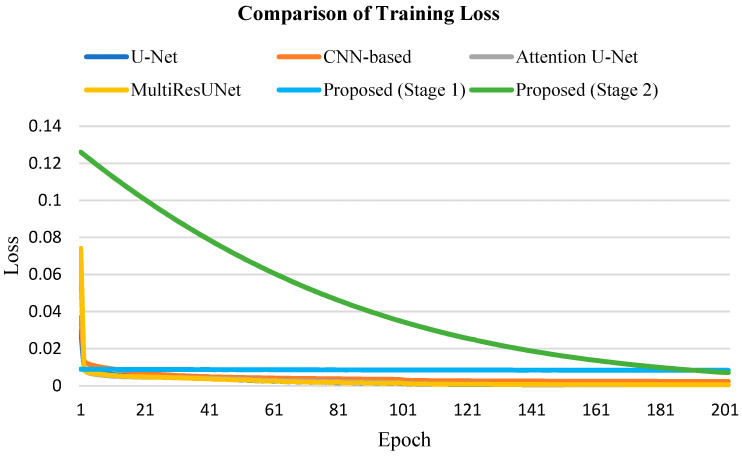
Comparison of training loss among different models.

**Figure 7 jcm-10-04760-f007:**
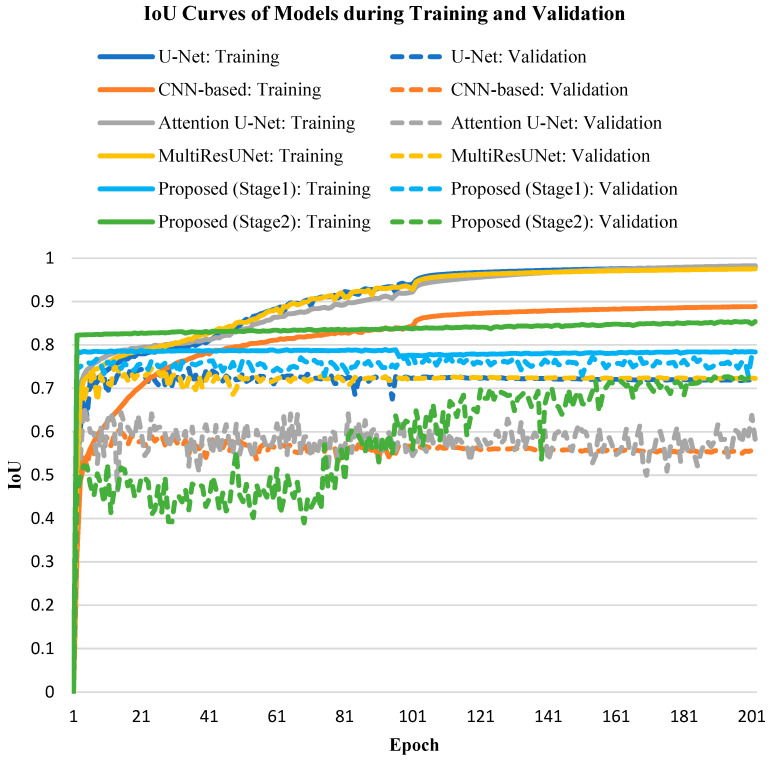
The curves of IoU during training and validation processes of different models: U-Net [9], CNN-based [10], Attention U-Net [11], MultiResUNet [12], and the proposed method (stage1 and stage2).

**Figure 8 jcm-10-04760-f008:**
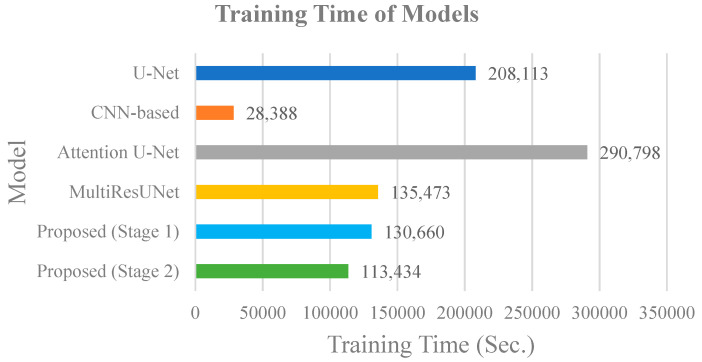
Comparison of training time costs of different models.

**Table 1 jcm-10-04760-t001:** Performance comparison of the proposed method and other methods.

Model	Number of Correctly Predicted Images (B)	Number of Error Predicted Images (T–B)	Accuracy (A)	Mean IoU
U-Net [9]	236	72	0.766	0.719
CNN-based [10]	41	267	0.133	0.555
Attention U-Net [11]	95	213	0.308	0.582
MultiResUNet [12]	257	51	0.834	0.723
Proposed	289	19	0.938	0.771

## Data Availability

Restrictions apply to the availability of these data. Data was obtained from Department of Radiology, Asia University Hospital and are available from Yi-Chi Huang with the permission of Asia University Hospital.

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
