# Peer review of "Automatic Segmentation of Specific Intervertebral Discs through a Two-Stage MultiResUNet Model"

_jcm, 2021, doi:10.3390/jcm10204760_

Round 1

Reviewer 1 Report

Summary:

This manuscript reported a study about a two-stage MultiResUNet model to segment the required specific intervertebral discs. In the first stage, the method uses a small range of MRI images for a part of target segmentation, and then extracts the distance feature using the distance transform from the segmentation results. In the second stage, the original MRI image combined with the distance maps is used to train the Multi-ResUNet to segment the remaining target. The results showed that the this model significantly improved segmentation performance with the segmentation accuracy to be about 94%, which is an increase of about 10% compared with the results of the MultiResUNet model. However, the mean IoU is only increased by about 5%. The Multi-ResUNet also needed a longer training time. The authors concluded that the segmentation errors and misalignments of specific intervertebral discs are greatly reduced, and the segmentation accuracy is increased to about 94%. They expected that the segmentation performance will definitely continue to improve with the improvement of the segmentation network architecture, more training data can be added or a loss function that is more suitable for intervertebral disc segmentation can be used.

Comments

This manuscript reported a new model with a two-stage MultiResUNet model to segment the required specific intervertebral discs. This design improved the accuracy of performance. As the mentioned by the authors in line 290, “there is still a long way to go before practical applications”. Some issues needed to be addressed.

  1. Lines 119-120, For the first stage, the images were manually marked three intervertebral discs between the lumbar spine and the sacrum by the physician, i.e. L1/L2, L4/L5 and L5/S1. However, how did the authors determined the correct level of these levels. In the clinical practice, some people have lumbar sacrization (in total 4 lumbar vertebrae), whereas some have sacrum lumbarization (in total 6 lumbar vertebrae). Usually, the correct level of lumbar spine can be determined by ribs. However, MRI images may not showed ribs for determination.
  2. Lines 184-185. “… in the second stage, distance features are added to modify the training data to help identification. The distance transform is used to compute the distance features of the lower two intervertebral discs.” However, the vertebral body has a cylinder shape and oriented with a lordosis curve. Therefore the so-called distance may change with different section of image. For this application, how did the authors define the distance? How did the authors select the “perfect” image for training? How about the precision errors of these results?
  3. Some patient had spondylolisthesis or obliterated vertebral disc spaces. The authors may need to include some image for training.
  4. The longer training time may present as a disadvantage for new model. Can the authors improve such issue?

Reviewer 2 Report

Introduction:

Avoid common language like "etc.".

"Backache has a great relationship with the degeneration of the spine." - This is unscientific.

Line 28-35 need to be rewritten completely. Also, the final sentences lack references.

line 54-66: Where does this lead the reader? Shorten or exclude.

line 67-86: Shorten substantially.

line 86-105: Shorten as well.

"We found that the segmentation results are not good and the segmentation errors often occur." This sentence cannot be serious.

Materials and Methods

Did you at first calculate the needed number of samples for a sufficient effect size? How were statistics made?

The technical description is far too detailed for the readership of JCM and a lot of it should go into supplementary material.

Results

Please summarize the results section into brief descriptions of your achievements. A lot of the figures can be summarized into larger ones. This way, the redundancy can be reduced.

Fig. 6 -8: If you are proposing a figure with Excel, at least get rid of the major grid, adapt the font and size, the color and create a new legend.

Round 2

Reviewer 1 Report

This revision manuscript reported a study about automatic segmentation of specific intervertebral discs through a two-stage MultiResUNet mode. The current manuscrtipt responded rather well to the comments. However, this study seemed rather preliminary.

  1. Lines 250-251" Because the number of images in these cases was small and this study focused on the segmentation and measurement of intervertebral discs, we did not classify any spinal diseases in the image data. Therefore, such a study took many assumption and that may stay a long distance from clinical application. The research design is always important for approaching further clinical significance. The authors may need to improve it.
  2. No section of discussion was included about the potential application and limitation. The further reseacrh direction should be also included for the refinement of such research. 
  3. In real clinical practice, many patients had previous spine surgery, including disectomy, instrumaentation, if any. Is this model applicable for their analysis?

Reviewer 2 Report

The introduction has been substantially improved.

Please rewrite the sentence in line 105: "were not good".

I am still missing a concise translational statement in the end of the abstract and conclusion.

The figures 6-8 have been substantially improved.
